# The Impact of Residents' Online Consumption on Offline Consumption—An Ordered Probit Semi-Parametric Estimation Method

**Xianjin Tu [1], Victor Shi [2], Ming Zhang [3],\* and Gangwu Lv [4],\***

1.  School of Economics and Management, Chongqing Normal University, Chongqing 400047, China; navytzk@163.com
2.  Lazaridis School of Business and Economics, Wilfrid Laurier University, Waterloo, ON N2L 3C5, Canada; cshi@wlu.ca
3.  School of State Governance, Southwest University, Chongqing 400715, China
4.  College of Resources and Environment, Southwest University, Chongqing 400715, China
\*   Correspondence: zhming523@swu.edu.cn (M.Z.); skclgw@swu.edu.cn (G.L.)

**Abstract:** Online consumption not only is an economic phenomenon, but also has a profound impact on offline consumption. Under this context, this article analyzes the mechanism of how they influence offline consumption and puts forward research hypotheses. China Household Financial Survey (CHFS) data and a semi-parametric ordered probit estimation method are used empirical tests. The results indicate that consumers with online consumption experience are very likely to consume again. The scale of online consumption not only drives the increase of overall consumption, but also promotes the growth of offline consumption via capital effect, complementarity effect, and psychologic effect. In general, online consumption and offline consumption have achieved integrated development.

**Keywords:** online consumption; offline consumption; ordered probit model; semi-parametric estimation

## 1. Introduction and Literature Review

The rapid development of Internet applications has changed the basic pattern of the consumer market, and the traditional restrictive barriers of physical "markets" have been knocked down and rebuilt using unrestricted virtual walls [1]. With the continuous improvement of network infrastructure and the rapid penetration of Internet technology innovation into the consumer market, online consumption has become an influential driving factor in the retail market. The scale of online consumption has also experienced exponential growth [2]. According to data from the National Bureau of Statistics of China, the consumption scale of China's online shopping has grown rapidly at a compound annual growth rate of 27.4% from 2015 to 2019, far exceeding the growth rate of 8.1% of the total retail sales in the same period. The Internet technology improves the consumer environment and brings convenience and benefits to consumers. However, in the context of China's economic landscape, which intertwines with multiple images such as economic growth shifts, structural adjustment pains, and old and new kinetic energy conversions, can the emergence of online consumption be integrated with offline consumption and even become a vital power for stimulating household consumption? If so, what is the path of online consumption affecting residents' offline consumption or even total consumption? Is online consumption an alternative or complementary effect to offline consumption? The answer to these questions has positive meaning with the formulation of relevant policies about releasing consumer consumption potential.

Accompanied with the increasing importance of domestic demand to whole economic growth in China, academia has paid more attention to this new consumption model, which is the online consumption. The existing research focuses on two aspects:

The first one is the analysis of the factors affecting consumers' online consumption. Some positive factors affecting residents' online consumption are found, such as loyalty [3,4], perceived value [5], trust [6], education and income [7], credit [8], herd behavior [9,10], and the way of buying [11]. On the contrary, factors that inhibit consumers' online consumption also exist, such as age [12], cost [13,14], risk [15], etc. In addition, some scholars also find several factors that have a more complicated impact on consumers' online consumption. Taking gender factors as an example, Kim et al. [16] and Davis et al. [17] argue that males tend to be more active toward online shopping, but Sener and Reeder [18] found that females shop online more frequently than males. Lian and Yen [19] and Lee et al. [7] even find that the influence of gender on online consumption is minimal or absent. There are still other factors affecting consumers' online consumption, including consumption motivation [20–22], values and lifestyles [23], and so on.

The second is the study of the relationship between online and offline consumption. There has been a long-standing debate on the relationship between online and offline consumption in academia [24], and three main viewpoints are mentioned. Firstly, online and offline consumption are alternatives. Online consumption has a crowding out effect on offline consumption. Online consumption occupies consumers' money and time, thereby reducing the likelihood of offline consumption [25–30]. Secondly, online consumption and offline consumption are complementary. Online consumption does not only inhibit offline consumption; on the contrary, online consumption will stimulate offline consumption. The reason is that great online shopping experiences can encourage offline consumption [31–34]. Thirdly, the relationship between online and offline consumption is more complicated. The relationship between them is not simply alternative or complementary [35]. It may be complementary in the short term but alternative in the long term [36], or it may also be a short-term alternative and long-term complementary relationship [37]. It is also possible that there may be no relationship between them [38].

Although the existing literature provides a lot of research on the relationship between online and offline consumption, it does not analyze the influence mechanism carefully and does not give us empirical evidence based on large-scale survey data. This study makes three main contributions to the literature, which is also the novelty of our paper. Firstly, compared with existing literature which mainly pays attention to the impact of online consumption experience, the study not only analyzes the impact of online consumption experience on offline consumption, but also tests the effect of scale of online consumption on scale of offline consumption. Secondly, based on traditional demand function and stimulus-organism-response theory, a theoretical analysis framework of how consumers' online consumption behavior has an impact on online re-consumption and offline consumption is constructed. Thirdly, using large-scale, nationwide micro-data (China Household Finance Survey, CHFS) to match the dataset, we employ a newly developed semi-parametric estimation methodology of the ordered probit model, which can get a uniform estimate even if we cannot measure the dependent variable precisely.

The rest of the paper is arranged as follows. Section 2 develops the research hypotheses about the possible impacts of online consumption on offline consumption. Section 3 presents the specific research design and introduces the empirical models, variables, and data. Section 4 presents the empirical tests and their results. Finally, Section 5 concludes the paper and discusses the research results.

## 2. Research Hypothesis

For a long time, due to objective factors such as time, space, and information, consumers have only been able to conduct personalized transactions in the real trading context. However, with the rapid development of the Internet economy and the proliferation of technology applications, shopping channels and the environment have undergone tremendous

changes, from the traditional single offline consumption to both offline and online. Online consumption is not only a habit for consumers but has gradually become a new way of life. The Internet has played an increasingly significant role in online consumption, and the popularity of Internet applications has changed consumers' values and behaviors [39–42]. Consumers are more and more tolerant of online consumption [33] and show a strong interest in online consumption.

The wide application of the Internet allows online consumption, but consumers' choice of online consumption will be affected by many factors. In general, there are three main categories: consumer factors, intermedia factors, and merchant factors. From the consumer's point of view, offline shopping is becoming a fashion and trend, and the consumer will be gradually influenced by other people, thus shopping via the Internet. Furthermore, the emerging new technologies and products have also produced innovative diffusion effects on consumer behavior decisions. In this case, consumers often tend to experience new products or services via online shopping [43]. What's more, online consumption has a cost advantage over offline consumption [12], which can get the largest consumer surplus with the least cost. When it comes to intermediary factors, Shankar et al. [44] believe that consumers' trust in online trading includes not only the trading subject, but also a range of technical support facilities including payment platforms, logistics services, etc. Online trading is more risky than offline trading, and traders' moral hazard and adverse selection are more serious [8], but the generation of Internet reputation mechanism can effectively overcome the market failure caused by information asymmetry [45,46]. The word-of-mouth effect produced by the feedback makes online transactions possible. If we take merchant factors into consideration, with the intensification of commercial competition, merchants have increased their shopping incentives for consumers, and various preferential policies have created a great temptation for consumers.

To some extent, the rapid development of the Internet has resulted in changes in consumer purchase behavior and repeated purchase intentions [23,47].

## 2.1. Online Consumption Experience and Household Consumption

Based on stimulus-organism-response theory [48,49], the existence of perceived value influences and determines the possibility of consumers' repeated purchase behavior [50]. The perceived value of consumers is the core structure and foundation of all exchange relationships and activities [47] and is a key driver of customer repurchase intentions [51]. In other words, through online consumption, in addition to obtaining physical product quality or price benefits, consumers can also obtain other valuable services that are attached to the physical attributes of products, such as social achievement, good customer service, convenient navigation, clear regulations, and comprehensive after-sales service, which are all part of the online shopping experience [52]. Compared with offline consumption, online shopping provides a bilateral platform environment. In the bilateral platform environment, there are a large number of customers who need goods and services and a large number of providers of goods and services. The formation of transactions in a bilateral platform often requires buyers and sellers to conduct frequent communication. A standardized and friendly communication experience enhances the customer's shopping experience. According to the stimulus-organism-response theory [48,49], consumers' purchasing decisions not only depend on the product itself, but also depend on the shopping experience, which can affect a consumer's psychological state, which in turn affects consumer behavior. Once a consumer has a sense of satisfaction from online shopping, it can positively stimulate the consumer's perceived value, thereby stimulating customers to repurchase [53]. Importantly, brand loyalty can be quickly established through a good shopping experience and high perceived value, and the product or service will become a reference point for the consumer to repurchase, further increasing the likelihood of re-consumption [3,4]. Therefore, the following research hypothesis of this paper is proposed:



**Hypothesis 1 (H1):** *People with online consumption experience are more likely to consume again. That is, online consumption experience is conducive to increasing people's consumption expenditure.*

*2.2. Online Consumption Scale and Household Consumption*

The consumer is more concerned with finding a balance between convenience, price, and service [54]. To get this balance, a consumer's shopping scene gradually extends from offline to online. However, online consumption is not separate from offline consumption, but also affects offline consumption through capital effect, complementarity effect, and psychologic effect. (1) Capital effect. Online consumption often has a price advantage compared to offline shopping [12]. Compared with the offline market, the e-commerce market has broken through the limitation of space, and the marginal cost of increasing the market size is basically zero [55]. The development of the Internet has also weakened the information asymmetry, and the transparency of the information and the high efficiency of the delivery of goods reduce the ability of merchants to control the price, which intensifies the market competition among enterprises. This competition is transmitted from the online market to the whole retail market (online and offline), and merchants cannot but provide a low price to capture the consumer [12]. A low price can effectively improve consumers' budget constraints. According to traditional demand function theory, the added budget can be used for offline consumption, which in turn promotes the release of offline consumption potential. (2) Complementary effect. Based on traditional demand function theory, the demand for one commodity is affected by the purchase of its substitutes. If the consumption of its substitutes increases, the purchase of this commodity will also increase. If the online commodity which people buy needs to be matched with another offline commodity, online shopping will drive offline shopping. (3) Psychologic effect. According to stimulus-organism-response theory [48,49], a good online shopping experience can exert a high perceived value, thereby stimulating customers to re-consume. If the demand for re-consumption is more urgent because of the good online consumer experience, it will make consumers impatient and rush to consume [56]. Because online shopping spends a lot of time on transportation, consumers tend to seek offline consumption. Therefore, the following research hypotheses are proposed:

**Hypothesis 2 (H2):** *The scale of online consumption has a pull effect on total consumption expenditure. That is, online consumption expenditure is conducive to increasing overall consumption.*

**Hypothesis 3 (H3):** *The scale of online consumption has a promoting effect on offline consumption. That is, online consumption expenditure is conducive to increasing offline consumption.*

**3. Model Design, Variable Selection, and Data Processing**

*3.1. Model Design*

The paper focuses on the impact of online consumption on consumption expenditure. Considering that the survey data is based on discrete data and there are hidden variables that cannot be observed directly, this paper uses the ordered probit model to empirically analyze the impact of online consumption on consumer behavior. In general, if we want to get the precise stratification of the dependent variable by 1–10, the ordered-probit model may have a reliability deviance. Thus, the paper divides the individual consumption expenditure into low (less than 20%), medium (20–80%), and high (more than 80%) consumer groups according to the national statistics bureau's classification of low, medium, and high income:

$$s^* = \begin{cases} 1 & s < 20\% \\ 2 & 20\% \leq s < 80\% \\ 3 & s \geq 80\% \end{cases} \tag{1}$$

Thus, using Equation (1), $s$ is re-divided into three mutually non-overlapping intervals, and we get a new variable $s^*$. Furthermore, we normalize the dependent variable to obtain a new variable $s'$ and establish the following relationship:

$$s^* = \begin{cases} 1 & s' < \zeta_1 \\ 2 & \zeta_1 \leq s' < \zeta_2 \\ 3 & s' \geq \zeta_2 \end{cases} \tag{2}$$

Next, the probability that a certain value corresponds to $s^*$ is:

$$pr[s^* = j] = \begin{cases} F(\zeta_1 - x_i'\beta) & j = 1 \\ F(\zeta_2 - x_i'\beta) - F(\zeta_1 - x_i'\beta) & j = 2 \\ 1 - F(\zeta_2 - x_i'\beta) & j = 3 \end{cases} \tag{3}$$

Among them, $F(\bullet)$ follows the normal distribution, $\zeta_1$ and $\zeta_2$ are the new interval division value. $x$ represents the explanatory variables, including net income, assets, etc., and $\beta$ represents the corresponding estimation coefficients. Next, we use $s^*$ as the explanatory variable to build an ordered probit model, and the log-likelihood function of this model is:

$$\ln L(\beta, \zeta_1, \zeta_2, \zeta_3) = \sum_{i=1}^{n} \sum_{j=1}^{3} 1\{s^* = j\} \ln[F(\zeta_{j+1} - x_i'\beta) - F(\zeta_j - x_i'\beta)] \tag{4}$$

In Equation (4), $1\{\bullet\}$ represents the characteristic function. When the condition in parentheses is satisfied, it is 1, otherwise, it is 0. By maximizing the log-likelihood function, the coefficient $\beta$ and parameters $\zeta_1$ and $\zeta_2$ can be estimated. However, the commonly used ordered probit model sets the residual $\varepsilon$ as the standard normal distribution when estimating the coefficient $\beta$, which is obviously hard to reach in practice. In this regard, Stewart [57] proposed that the semi-parametric method can be used to correct. Assuming a function distribution that is not known beforehand, the density function $\varepsilon$ is approximated by the Hermit sequence $f_k(\varepsilon) = 1/\alpha * \left( \sum_{\rho=0}^{k} \zeta_s \varepsilon^2 \right)^2 \prod(\varepsilon)$, and the cumulative distribution function $F_k(\bullet)$ is substituted for the likelihood function $F(\bullet)$. Thus, an estimation of the parameter $\beta$ is obtained.

### 3.2. Variable Selection

The focus of the research is whether online consumption has an impact on overall consumption and offline consumption. Specifically, we will take the impact of online consumption experience and online consumption scale into consideration together. The major variables are as follows:

(1) Consumption scale. It mainly refers to residents' daily consumption expenditure levels, including food, clothing, housing, daily necessities and services, transportation and communication, education, culture and entertainment, health care, and other consumer expenditures. Specifically, when we estimate the scale of consumption, it excludes online consumption from total consumption expenditure.

(2) Online consumption experience. It mainly refers to whether residents consume on the Internet. If they do, the value is 1, otherwise, the value is 0.

(3) Other control variables: (a) Income. It is the annual income obtained from consumption, including wage income, operating income, property income, and transfer income. (b) Assets. It includes financial assets (cash, deposits, stocks, securities, funds, gold, etc.), real estate, various durable goods, etc. (c) Age. It is assigned as the respondent's age in 2013. (d) Gender. The value of the variable is 1 for male and 0 for female. (e) Marriage. If the respondent is married, it is assigned as 1. Otherwise, it is assigned as 0. (f) Education level. According to the education that householders received, 1–9 respectively represent non-educated, primary school, junior high school,

senior high school, technical secondary school/vocational school, bachelor degree, master degree, doctoral degree. (g) Political status. According to householder's political status, 1–4 respectively represent Communist Youth League member, masses, party member of CPC, democratic parties. (h) Household registration. The value of the variable is 1 for registered urban residents and 0 for registered rural residents. (i) Happiness index. It reflects citizens' mental needs. The higher the happiness index is, the higher the autonomy of customers is; thus, interaction technology driven by autonomous motivation will enable consumers to experience higher choice [58]. The responses from "Are you happy?" in the questionnaire are ranked from 1 to 5, and it respectively suggests "very unhappy", "unhappy", "so so", "happy", "very happy". (j) Health status. It is ranked from 1 to 5 based on responses from the question "How is your health condition?", namely, "bad", "so so", "good", "quite good", "very good". (k) Risk attitude. It is used to demonstrate citizens' expectations for future consumption decision-making and to reflect to what extent consumers are willing to make decisions based on their own willingness. The responses from the question "What type of investment would you choose if you own a sum of money?" were ranked 1–5 and respectively represent "not willing to take any risks", "slightly low risk with slightly low return", "average risk with average return", "slightly high risk with slightly high return", "high risk with high return". (l) Family size. It is presented as the number of family members living together. (m) Social capital. It is presented as the number of siblings living who do not live with the residents. (*n*) Distance. It is used to show the time cost of residents' consumption decisions, and it is presented as the time that residents spend on traveling to the downtown.

### 3.3. Data Processing

The paper focuses on the spillover effect of online consumption on household consumption. All the required data are from the 2013 China Household Finance Survey (CHFS). After screening the survey data and deleting the sample with an income of 0, valid samples were taken respectively. By observing the descriptive statistical results in Table 1, it can be seen that the online consumption experience is 0.2869, and the amount of online consumption is only 5.29 thousand yuan. Compared with a total consumption of 69.111 thousand yuan, it indicates that online consumer behavior needs further popularization. Compared with the whole sample, residents with online consumption experience have higher income, assets, happiness index, educational level, health status, risk attitude, etc. In addition, we can see gender and age differences in online consumption, and young people and women have higher enthusiasm for online consumption.

**Table 1.** Descriptive statistics of the main variables (number of observations: full sample 16,706, sub-sample 4549).

| Variable | Unit | Full Sample | | Sub-Sample | |
| --- | --- | --- | --- | --- | --- |
| | | Average | Standard Derivation | Average | Standard Derivation |
| Consumption | 10,000 yuan | 4.0158 | 5.7286 | 6.9111 | 8.9068 |
| Offline consumption | 10,000 yuan | — | — | 6.3821 | 8.5249 |
| Income | 10,000 yuan | 2.3088 | 11.4748 | 5.0830 | 16.0788 |
| Online consumption experience | | 0.2869 | 0.4523 | — | — |
| Online consumption scale | 10,000 yuan | — | — | 0.5290 | 1.0275 |
| Assets | 10,000 yuan | 16.6935 | 62.1113 | 36.1670 | 78.7046 |
| Happiness index | | 3.6241 | 0.8568 | 3.7534 | 0.8066 |

**Table 1.** *Cont.*

| Variable | Unit | Full Sample | | Sub-Sample | |
|---|---|---|---|---|---|
| | | Average | Standard Derivation | Average | Standard Derivation |
| Gender | | 0.5934 | 0.4912 | 0.5186 | 0.4997 |
| Age | | 47.9652 | 13.6108 | 39.5931 | 11.6519 |
| Education level | | 3.5434 | 1.7930 | 5.1451 | 1.7349 |
| Political status | | 2.3674 | 0.7461 | 2.5830 | 0.8577 |
| Household registration | | 0.3960 | 0.4891 | 0.7254 | 0.4463 |
| Marriage | | 0.9315 | 0.2527 | 0.8549 | 0.3522 |
| Health status | | 2.6750 | 1.2004 | 3.1205 | 1.1361 |
| Risk attitude | | 2.1070 | 1.2941 | 2.5779 | 1.2143 |
| Family size | | 3.6944 | 1.6132 | 3.3838 | 1.3054 |
| Social capital | | 2.7931 | 1.1459 | 2.8846 | 1.1350 |
| Distance | minute | 38.3490 | 37.5304 | 23.6316 | 24.3989 |

Further analysis of the total samples and sub-samples can be seen in Table 2. In terms of gender, male samples are slightly more than female ones. In addition, samples are mainly middle-aged people (78.82% of total sample and 91.04% of sub-sample). It can also be found that most samples feel happy (84.10% and 91.04%). In respect of risk-taking, most residents are non-risk-preferred (86.84% and 80.31%). As for family size, most samples are a small family, consisting of 2–4 persons (69.95% and 75.44%).

**Table 2.** Sample distribution.

| Variables | Content | Total-Sample | | Sub-Sample | |
|---|---|---|---|---|---|
| | | Sample Size | Percentage (%) | Sample Size | Percentage (%) |
| Gender | female | 6793 | 40.66 | 2190 | 48.14 |
| | male | 9913 | 59.34 | 2359 | 51.86 |
| Age | the young (18–44) | 297 | 1.79 | 187 | 4.12 |
| | the middle-aged (45–59) | 13,171 | 78.82 | 4140 | 91.04 |
| | the old (60 and above) | 3238 | 19.42 | 222 | 4.88 |
| Political Status | Communist Youth League member | 42 | 0.25 | 17 | 0.37 |
| | masses | 13,151 | 78.72 | 2966 | 65.20 |
| | party member of CPC | 2666 | 15.96 | 1103 | 24.25 |
| | democratic parties | 847 | 5.07 | 463 | 10.18 |
| Happiness Index | very unhappy | 214 | 1.28 | 28 | 0.62 |
| | unhappy | 1090 | 6.52 | 188 | 4.13 |
| | so so | 5898 | 35.30 | 1447 | 31.81 |
| | happy | 7064 | 42.28 | 2101 | 46.19 |
| | very happy | 2440 | 14.61 | 785 | 17.26 |

**Table 2.** *Cont.*

| Variables | Content | Total-Sample | | Sub-Sample | |
|---|---|---|---|---|---|
| | | Sample Size | Percentage (%) | Sample Size | Percentage (%) |
| Health Status | bad | 2661 | 15.93 | 235 | 5.17 |
| | so so | 6276 | 37.57 | 1427 | 31.37 |
| | good | 2976 | 17.81 | 1004 | 22.07 |
| | quite good | 3418 | 20.46 | 1321 | 29.04 |
| | very good | 1375 | 8.23 | 562 | 12.35 |
| Risk Attitude | not willing to take any risks | 7901 | 47.29 | 1145 | 25.17 |
| | low risk with low return | 2765 | 16.55 | 901 | 19.81 |
| | average risk with average return | 3843 | 23.00 | 1607 | 35.33 |
| | slightly high risk with slightly high return | 932 | 5.58 | 528 | 11.61 |
| | high risk with high return | 1078 | 6.45 | 361 | 7.94 |
| Social Capital | 1 sibling | 2809 | 16.81 | 689 | 15.15 |
| | 2 siblings | 4626 | 27.69 | 1155 | 25.39 |
| | 3 siblings | 2484 | 14.87 | 697 | 15.32 |
| | 4 siblings | 6787 | 40.63 | 2008 | 44.14 |
| Family Size | 1 person | 574 | 3.44 | 294 | 6.46 |
| | 2 persons | 3139 | 18.79 | 492 | 10.82 |
| | 3 persons | 5205 | 31.16 | 2166 | 47.61 |
| | 4 persons | 3341 | 20.00 | 774 | 17.01 |
| | 5 persons | 2357 | 14.11 | 572 | 12.57 |
| | 6 persons | 1303 | 7.80 | 177 | 3.89 |
| | 7 persons and above | 787 | 4.72 | 74 | 1.62 |

## 4. Online Consumption and Household Consumption: Empirical and Analysis

*4.1. The Impact of Online Consumption Experience on the Scale of Household Consumption*

In order to accurately describe the impact of online consumption behavior on offline consumption, based on the analysis of the consumers' online consumption behavior framework above, the paper applies the ordered probit model to empirically test the impact of online consumption experience on residents' consumption at first. Generally, the semi-parametric method is used to estimate with different orders of $k$. It is proved that the empirical test is equivalent to the ordered probit model when $k = 2$ [57], so it can be said that the starting point of semi-parametric estimation is $k = 3$. In addition, the likelihood ratio test (LR) can be used to judge whether the semi-parametric estimation is necessary and the choice of $k$ value. The LR test here mainly includes two types: one is an ordinary LR (OP) test for checking whether the semi-parametric estimation is necessary or not, and the other is the LR ($k$-1) test for determining the ($k$-1) order of the $k$ value.

The LR test results in Table 3 show that the semi-parametric test results are significantly different from the parameter estimation results, and the semi-parametric estimation is the optimal choice. In addition, semi-parameter estimation is more suitable when $k = 3$. Therefore, $k = 3$ is used for semi-parametric estimation of the ordered probit model. In Table 4, the parameter estimation and the semi-parametric estimation results are almost

the same in the significance level and the influence direction, but there are significant differences in the degree of influence, which indirectly confirms the reliability of the LR test results and provides sufficient statistical support for using the semi-parametric estimation.

**Table 3.** LR test of models corresponding to different *k*.

| *k* | Log Likelihood Value | LR OP | Degree of Freedom | *p* Value | LR (*k*-1) | *p* Value |
|---|---|---|---|---|---|---|
| OP | −12,486.418 | | | | | |
| 3 | −12,455.521 | 61.7927 | 1 | 0.0000 | 61.79 | 0.0000 |
| 4 | −12,404.083 | 164.6686 | 2 | 0.0000 | 102.88 | 0.0000 |
| 5 | −12,403.944 | 164.9474 | 3 | 0.0000 | 0.28 | 0.5975 |
| 6 | −12,389.919 | 192.9968 | 4 | 0.0000 | 28.05 | 0.0000 |

Note: The null hypothesis of the LR (OP) test is *k* = 2, and the alternative hypotheses are *k* = 3, 4, 5, and 6 in extended models, respectively. The null hypothesis of the LR test in the latter two columns is the (*k*-1) order extended model, the alternative hypothesis is the k order extended model, and the degrees of freedom of the test are all 1.

**Table 4.** Ordered probit model (OP) parameters and semi-parametric estimation results.

| | Parameter Estimation | | Semiparametric Estimation | |
|---|---|---|---|---|
| | Marginal Effect | Standard Error | Marginal Effect | Standard Error |
| Income | 0.0060 *** | 0.0009 | 0.0079 *** | 0.0010 |
| Online consumption experience | 0.5481 *** | 0.0263 | 0.6887 *** | 0.0411 |
| Assets | 0.0021 *** | 0.0002 | 0.0020 *** | 0.0002 |
| Happiness index | 0.0667 *** | 0.0115 | 0.0842 *** | 0.0131 |
| Gender | −0.0898 *** | 0.0201 | −0.1156 *** | 0.0238 |
| Age | −0.0168 *** | 0.0009 | −0.0182 *** | 0.0012 |
| Education level | 0.1425 *** | 0.0083 | 0.1813 *** | 0.0113 |
| Political status | 0.0526 *** | 0.0140 | 0.0624 *** | 0.0163 |
| Household registration | 0.5915 *** | 0.0262 | 0.7002 *** | 0.0379 |
| Marriage | 0.5231 *** | 0.0424 | 0.6588 *** | 0.0537 |
| Health status | 0.0716 *** | 0.0087 | 0.0862 *** | 0.0103 |
| Risk attitude | 0.0394 *** | 0.0077 | 0.0512 *** | 0.0090 |
| Family size | 0.1460 *** | 0.0064 | 0.1666 *** | 0.0082 |
| Social capital | 0.0578 *** | 0.0083 | 0.0715 *** | 0.0098 |
| Distance | −0.0031 *** | 0.0003 | −0.0036 *** | 0.0004 |
| Wald $\chi^2$ value | | | 1027.12 | |
| *p* value | | | 0.000 | |
| Likelihood value | | | −12,455.521 | |
| Skewness | | | 0.3389 | |
| Kurtosis | | | 3.0475 | |
| Standard deviation | | | 1.1960 | |

Note: *** indicate significant at 1%.

According to the semi-parametric estimation results in Table 4, the marginal effect of online consumption experience on household consumption is 0.6887, and it passes the significance test at the 1% level, which indicates that past online consumption experience is also the driving force for consumers to increase consumption again [15]. The empirical conclusion is consistent with the theoretical analysis, thus validating research hypothesis 1. This is because consumers get a good experience or high perceived value in the online consumption process, which evokes the desire to re-consume again. Online consumption can help consumers have a good experience via social achievement, good customer service, convenient navigation, clear regulations, and comprehensive after-sales service, so they tend to re-consume. The result is consistent with Zhang et al. [33]. Zhang et al. [33] find that when the consumer has a good consumption experience from online shopping, it will become the initial driving force for the sustainable development of online consumption together with offline consumption. The consumption effect of income is significantly positive, indicating that income is still a positive variable affecting consumer behavior decision-making, which is consistent with classical consumption theory. Although the consumption effect of assets is significant, it is weaker than that of income variables. This is similar to Yan and Chen [59]. Using China's monthly macro-economic data, Yan and Chen [59] conclude that, although there is increasing impact of household assets on consumption, income is still the main factor which drives consumption. By comparing the effects of online consumption, income, and assets, the impact of income and assets is significantly less than online consumption experience, which may mean that online consumption will be a new driving force for consumption.

According to the estimation results, it cannot be ignored that other control variables also have a significant effect on household consumption. The coefficient of happiness index is positive, indicating that optimistic emotions are good to arouse residents' desire for consumption. The coefficient of gender is negative, indicating the existence of behavioral differences between males and females, and females are the main group of online consumption. The coefficient of age is negative, indicating that young people are more motivated to consume. The coefficient of education level is positive, showing that the higher the educational level, the greater the consumption expenditure. The political status is positive, indicating that it will also affect consumption. The coefficient of household registration is positive, indicating the consumption difference is large between urban and rural residents. The coefficient of marriage is positive, indicating that married residents have more consumption expenditure than unmarried residents. The coefficient of health status is positive, which means health is important in promoting household consumption. The risk attitude is positive, indicating that risk appetite helps to increase the likelihood of consumption in the context of uncertainty. The coefficient of family size is positive, and it is due to rigid consumption of family size. The coefficient of social capital is positive, and developed social networks can provide consumers with funds, information, and other support, which increases consumption. The coefficient of distance is negative, because the geographical distance is a stumbling block to residents' consumption.

### 4.2. The Impact of Online Consumption on Household Consumption

The previous section shows that online consumption experience is conducive to increasing consumer expenditure. In order to match the research with real life, we need to further explore the impact of online consumption scale on total consumer expenditure and offline consumption expenditure. In view of the consistency of the research conclusions, we still use the semi-parametric probit model for empirical testing. We selected 4549 valid samples which have online consumption experience. The LR test shows it is appropriate for the use of semi-parameters for estimation, where the $k$ value of the total consumer expenditure model and offline consumption expenditure is 3 and 4, respectively.

The semi-parametric estimation results in Table 5 show that the marginal effect of online consumption scale on total consumer expenditure is 0.3754, and it is highly significant, which indicates that the scale of online consumption directly promotes the growth of total

consumption expenditure, which is consistent with the previous theoretical analysis, and hypothesis 2 is validated. Secondly, the marginal effect of online consumption scale on offline consumption is also significantly positive, which shows that online consumption, as a supplement to the offline market, does not crowd out offline consumption, but further increases offline consumption, which verifies hypothesis 3. The online market has brought positive externality to the offline market, and the expansion of online consumption creates fierce competition between the online market and offline market, thereby increasing market efficiency. The conclusion also brings more realistic thinking: online consumption and offline consumption can achieve integrated development through strategic cooperation. In the Internet economic era, it provides a variety of preferential subsidies and can help people get more consumer surplus. Consumers can get goods at a lower price, and the saved budgets can be used for consumption in the future [60]. Of course, it can be said that online shopping greatly increases consumption stickiness. This shows that online and offline consumption can achieve sustainable and integrated development through strategic cooperation rather than an inverse relationship.

**Table 5.** Semi-parametric estimation results.

| | Total Consumption | | Offline Consumption | |
|---|---|---|---|---|
| | **Marginal Effect** | **Standard Error** | **Marginal Effect** | **Standard Error** |
| Income | 0.0115 *** | 0.0013 | 0.0124 *** | 0.0014 |
| Online consumption | 0.3754 *** | 0.0254 | 0.1362 *** | 0.0200 |
| Asset | 0.0022 *** | 0.0002 | 0.0020 *** | 0.0002 |
| Happiness index | 0.0271 | 0.0189 | 0.0290 | 0.0177 |
| Gender | −0.1450 *** | 0.0310 | −0.1179 *** | 0.0291 |
| Age | 0.0076 *** | 0.0016 | 0.0064 *** | 0.0015 |
| Education level | 0.1081 *** | 0.0113 | 0.0955 *** | 0.0121 |
| Political status | 0.0129 | 0.0184 | 0.0049 | 0.0170 |
| Household registration | 0.2243 *** | 0.0407 | 0.2140 *** | 0.0405 |
| Marriage | 0.3279 *** | 0.0534 | 0.3289 *** | 0.0564 |
| Health | 0.0444 *** | 0.0139 | 0.0469 *** | 0.0136 |
| Risk attitude | 0.0860 *** | 0.0128 | 0.0800 *** | 0.0128 |
| Family size | 0.1329 *** | 0.0122 | 0.1286 *** | 0.0142 |
| Social capital | 0.0389 *** | 0.0133 | 0.0287 ** | 0.0123 |
| Distance | 0.0004 | 0.0006 | 0.0000 | 0.0006 |
| Wald $\chi^2$ value | 1016.12 | | 258.16 | |
| $p$ value | 0.000 | | 0.000 | |
| Likelihood value | −3689.9222 | | −3882.0290 | |
| Skewness | 0.0522 | | −0.5778 | |
| Kurtosis | 4.61596 | | 6.9489 | |
| Standard deviation | 0.8890 | | 1.2449 | |

Note: *** and ** indicate significant at 1% and 5% respectively.

*4.3. Robust Test of the Impact of Online Consumption on Consumer Behavior*

The empirical results above show that online consumption experience helps to increase consumption scale, and the scale of online consumption stimulates total expenditure and offline consumption. However, the reliability of the empirical results may be affected by

different sample selections. Based on this consideration, it is necessary to re-select samples to test the reliability of the empirical results. Therefore, based on the original samples, the lowest-income sample (5%) and the highest-income sample (5%) were deleted respectively, and we will focus on whether the impact of online consumption on residents' consumption is robust or not. In terms of model selection, the parametric and semi-parametric LR test results show that the semi-parametric test is more suitable for the empirical model. The optimal choice of $k$ is basically consistent with the previous one. In Table 6, the marginal effect of online consumption experience on household consumption is 0.6887 and 0.7519, respectively, the marginal effect of online consumption scale on total consumption expenditure is 0.3754 and 0.4606, and the marginal effect of online consumption on offline consumption is 0.1362 and 0.1220, respectively, and there is no significant change. Thus, the empirical results about the effect of online consumption are robust, which verifies research Hypotheses 1, 2, and 3 again.

**Table 6.** Robustness test.

| | Online Consumption Experience | | Online Consumption Scale | | | |
| --- | --- | --- | --- | --- | --- | --- |
| | | | Total Consumption | | Offline Consumption | |
| | Marginal Effect | Standard Error | Marginal Effect | Standard Error | Marginal Effect | Standard Error |
| Income | 0.1445 *** | 0.0099 | 0.0786 *** | 0.0132 | 0.0686 *** | 0.0125 |
| Online consumption | 0.7519 *** | 0.0480 | 0.4606 *** | 0.0725 | 0.1220 *** | 0.0294 |
| Assets | 0.0017 *** | 0.0002 | 0.0023 *** | 0.0004 | 0.0019 *** | 0.0004 |
| Happiness index | 0.0698 *** | 0.0156 | 0.0259 | 0.0255 | 0.0202 | 0.0228 |
| Gender | −0.2231 *** | 0.0290 | −0.3015 *** | 0.0567 | −0.2385 *** | 0.0546 |
| Age | −0.0170 *** | 0.0012 | 0.0119 *** | 0.0027 | 0.0092 *** | 0.0022 |
| Education level | 0.1628 *** | 0.0135 | 0.0883 *** | 0.0197 | 0.0695 *** | 0.0170 |
| Political status | 0.0811 *** | 0.0201 | 0.0155 | 0.0252 | −0.0045 | 0.0216 |
| Household registration | 0.8217 *** | 0.0401 | 0.3328 *** | 0.0709 | 0.2899 *** | 0.0669 |
| Marriage | 0.6719 *** | 0.0675 | 0.3114 *** | 0.0806 | 0.3057 *** | 0.0876 |
| Health | 0.0804 *** | 0.0123 | 0.0424 ** | 0.0198 | 0.0459 ** | 0.0185 |
| Risk attitude | 0.0394 *** | 0.0107 | 0.0810 *** | 0.0204 | 0.0655 *** | 0.0185 |
| Family size | 0.1876 *** | 0.0091 | 0.1805 *** | 0.0302 | 0.1632 *** | 0.0306 |
| Social capital | 0.0945 *** | 0.0118 | 0.0376 ** | 0.0183 | 0.0192 | 0.0155 |
| Distance | −0.0039 *** | 0.0004 | 0.0012 | 0.0009 | 0.0004 | 0.0007 |
| Wald $\chi^2$ | 1490.19 | | 53.97 | | 48.12 | |
| $p$ value | 0.000 | | 0.000 | | 0.000 | |
| Likelihood value | −11,080.9930 | | −3258.2815 | | −3442.2453 | |
| Skewness | 0.4309 | | 0.3599 | | −0.2371 | |
| Kurtosis | 2.6776 | | 3.2022 | | 5.9281 | |
| Standard deviation | 1.4251 | | 1.0965 | | 1.1759 | |

Note: *** and ** indicate significant at 1% and 5% respectively.

## 5. Conclusions and Policy Proposals

Online consumption has become a new choice for consumers. Therefore, it is vital to deeply analyze the internal logic of consumers' behavior choice with online consumption. Based on this, this paper constructs a theoretical framework for the impact of online consumption on consumers' behavior. Using the data from the 2013 China Household Finance Survey (CHFS), the prudent and reliable conclusion is obtained by using the semi-parametric ordered probit model. The results show that the online consumption experience is conducive to restarting consumption, and the scale of online consumption not only drives the increase of overall consumption, but also promotes the growth of offline consumption via capital effect, complementarity effect, and psychologic effect. In general, online consumption and offline consumption have achieved integrated development.

Our conclusions in this paper can enrich the academic literature on the impact of online consumption on offline consumption, and it can also provide insights for online consumption development in countries like China. Specifically, our empirical results lead to the following policy insights. Firstly, the government should actively promote the process of internetization, increase network coverage, allow residents to enjoy the convenience of the Internet, and further stimulate online consumption together with offline consumption. It is necessary to improve the online consumption experience, because it is the basis for online consumption to generate re-consumption and stimulate offline consumption. In the future, it is important to use Internet technology to accurately capture consumer needs and then improve consumer satisfaction with online consumption. Secondly, the government needs to improve e-commerce laws and regulations and build safe and reliable online shopping protection. The relevant laws and regulations must cover the full consumption chain of supply (merchants)-mediation (platform)-demand (consumers). Only this can dispel people's doubts about online shopping and achieve a steady growth of online consumption. Finally, the government should promote the development of e-commerce in small- and medium-sized cities and rural areas in China by improving Internet hardware and online shopping platform policy. In these areas, there is huge consumption potential. If the government improves the environment of online trading in these areas, it is conductive to exaggerating China's domestic demand.

Our study fills an important gap in the literature by investigating the impacts of online consumption on offline consumption. However, due to the limitations of data, there are still many aspects which could be future research directions. Firstly, this study takes the data of the China Household Finance Survey in 2013 as samples. If we get further years' data, we may be able to draw richer conclusions and policy recommendations. Secondly, online consumption can affect outline consumption via various mechanism. If we can obtain corresponding data, such as frequency/time of internet use and the place of internet access, we can test it and provide empirical evidence. Thirdly, there are lots of factors which will exert impact on households' consumption behavior; we may get more robust results if our models can control these variables. Fourthly, the online consumption scale regarding food, clothing, housing, and other commodities is different, and how online consumption of different commodities affects follow-up consumption is not discussed in this article. This is also the direction of future research.

**Author Contributions:** X.T. and V.S. conceived the main idea of the theoretical mechanism and manuscript drafting. M.Z. and G.L. did empirical analysis. All authors have read and agreed to the published version of the manuscript.

**Funding:** This research was funded by the Fundamental Research Funds for Central Universities (No. SWU1909017).

**Institutional Review Board Statement:** Not applicable.

**Informed Consent Statement:** Not applicable.

**Data Availability Statement:** Data available on request due to restrictions e.g., privacy or ethical. The data presented in this study are available on request from the corresponding author. The data are not publicly available as the data also forms part of an ongoing study.

**Conflicts of Interest:** The authors declare no conflict of interest.

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
