# Peer review of "The Impact of Residents’ Online Consumption on Offline Consumption—An Ordered Probit Semi-Parametric Estimation Method"

_sustainability, doi:10.3390/su131810047_

Round 1

Reviewer 1 Report

Comments and Suggestions for Authors The paper only proposes an empirical study not supported by a clear literature review. The research hypotheses are not derived from the literature but they are only related to the author/s perspective. More than this, the results are simple listed without a clear discussion able to support the readers in understanding the advancements in knowledge provided by the main text while implications and conclusions are completely underestimated. Finally a professional review of the language is strongly suggested for solving several typos and redundancies. In nutshell, the paper requires to be strongly revised before evaluating a possible publication.

Author Response

Thank you very much for your time and efforts reviewing our paper. You have provided a number of highly constructive and insightful suggestions and comments. Accordingly, we have revised and improved our paper significantly. Please see our detailed response below.

Reviewer 2 Report

After reviewing this revised and resubmitted manuscript I am satisfied the authors have addressed the referee comments. 

Minor issues:

  • page 2, row 47: "economdic" - it should be "economic";
  • page 2, row 68: "the second..." - it should be "The second..."
  • page 14, row 481: "this variables" - it should be "these variables"

Author Response

(The authors gave the same response as above.)

Reviewer 3 Report

The paper, “A New Driver of Sustainable Consumption: a Perspective of Online Consumption’s Integration with Offline Consumption”, addresses a research area interesting in consumption area. However, some aspects should be considered. For example, the Introduction section, the authors draw on several prior studies, but a much more critical literature analysis is needed to strengthen the paper’s argument and draw out the gaps they seek to address. Which gap(s) in extant studies are authors trying to address here?  That means, the introduction should clearly indicate the need for this paper in relation to extant research studies.

While the author(s) establish some links to some literature about several constructs, author(s) need to establish a more coherent framework for the overall paper and more recent literature and a theoretical perspective can be used to support the research hypotheses. A theory can be used to reinforced the hypotheses development. Our literatures, across several field and disciplines, is extensive on the constructs here studied, but strong theorizing remains scarce.  It is this strong theorizing that is needed to explain the phenomenon here proposed.

Your study is a purely empirical study.   Your hypotheses are correlational.   They associate one set of variables/constructs with another set.   You make no attempt to theorize from the results of your hypotheses testing.  Also, the paper needs to be present much stronger discussion and conclusion sections in order to offer value to the reader. Overall, the manuscript makes some interesting points, but the study aims to study some relations among the variables/constructs. As such, we have a better chance of making a contribution to theory, which is sought. You do not situate your study in a theoretical frame.

While you have made a valiant attempt to tackle these topics, I would argue that the data used from the final sample is “good” and the statistical techniques used are robust to support the empirical study and to achieve the main research objective. However, the results obtained should be discussed with more comparative previous!

The conclusions and implications could be extended, innovative and more contributions for public policy should also be presented. Some limitations and future agenda of this research field should be inserted!

Author Response

(The authors gave the same response as above.)

Round 2

Reviewer 1 Report

After the review the paper seems to be more clear and better structured. My congrats for your effort and for the result.

Reviewer 3 Report

The authors inserted my suggestions. In opinion the paper is more robust and it was improved.

This manuscript is a resubmission of an earlier submission. The following is a list of the peer review reports and author responses from that submission.

Round 1

Reviewer 1 Report

Thank you for the opportunity to review the study titled” A New Driver of Sustainable Consumption: a Perspective of Online
Consumption’s Integration with Offline Consumption”, which identifies an innovative perspective/methodology to study how online consumption can have an impact on offline consumption.

The article includes a review/description of this phenomenon and it is well informed, but... not complete, in particular more recent literature can be inserted. In addition, a theory can be proposed to support this study and hypotheses development.

In my opinion this paper focuses an interesting topic in sustainability area. The Introduction section is good, but in this section the authors could more clearly the gap and the main contributions of the study. Overall, the paper offers an interesting overview for this innovative topic and sustainability context that has not received in-depth attention in the scholarly debate, more precisely, about the impact that online consumption can have on offline consumption….

The methodology section can be improved. Please reinforce why you select these variables. The sample selection can also be more clearly explained. However, in Results section more discussion is necessary. There is at certain points but the arguments are unconvincing. The implications for theory and practice could be inserted, innovative and more contributions for practice should be presented. Some limitation of the study and suggestions for the future are also necessary.

Reviewer 2 Report

The paper is interesting and the empirical part is based on a good data set. This said some of the variables presented in Section 3.2. need a bit more explanation. If take for example, social capital, political status, risk attitude, the happiness index, are there any minimum and maximum values? Furthermore, in the case of the happiness index, what is the proportion of the people located at the extremes of the scale (very unhappy and very happy, respectively); household size (what is the most frequent size?); political status is interpreted in the paper in the discussion of the results as equivalent to social status?; concerning the age, there is no information either about the minimum age of the individuals included in the sample and it would be good to have also the proportion of the young people (18-35 years?); the variable gender, it is given the average but the proportion of men and women in the sample would be much more informative.  

The variable “distance” must be also explained in section 3.2. Otherwise one should go to the discussion of the results presented in Table 3 (line 398) to learn about it.

There is rather scarce information about the products purchased online (rows 287-288)  but it would be good to know what products/services are the most purchased by individuals. The author(s) is arguing that online/offline consumption could be either substitute, complementary or in a different relationship; this might very well depend on the type of products purchased online.

Last but not least, given that the analysis is dealing with online consumption, I wonder whether the author(s) considered including in the analysis  some variables related to the individuals’ behavior with respect to Internet access: frequency/time of internet use; the place of  internet access, etc. as they are expected to have an impact on the online consumption.

Minor issues:

  • Row 281: “thesis” is better to be changed by “research”, for example.
  • Row 377: … According, should be according
  • 398: the previous article, it should be the previous section

Reviewer 3 Report

I would really thank you the authors for this interesting reading. Unfortunately, it is not clear for me in which ways the paper contributes to the advancements in knowledge related to the domain of sustainability. Accordingly, it cannot be published in this journal.